# Approach to the Quantitative Diagnosis of Rolling Bearings Based on Optimized VMD and Lempel–Ziv Complexity under Varying Conditions

**DOI:** 10.3390/s23084044

**Published:** 2023-04-17

**Authors:** Haobo Wang, Tongguang Yang, Qingkai Han, Zhong Luo

**Affiliations:** 1School of Mechanical Engineering and Automation, Northeastern University, Shenyang 110819, China; 2Key Laboratory of Vibration and Control of Aero-Propulsion System Ministry of Education, Northeastern University, Shenyang 110819, China

**Keywords:** quantitative diagnosis, rolling bearing, optimal variational modal decomposition, Lempel–Ziv complexity

## Abstract

The quantitative diagnosis of rolling bearings is essential to automating maintenance decisions. Over recent years, Lempel–Ziv complexity (LZC) has been widely used for the quantitative assessment of mechanical failures as one of the most valuable indicators for detecting dynamic changes in nonlinear signals. However, LZC focuses on the binary conversion of 0–1 code, which can easily lose some effective information about the time series and cannot fully mine the fault characteristics. Additionally, the immunity of LZC to noise cannot be insured, and it is difficult to quantitatively characterize the fault signal under strong background noise. To overcome these limitations, a quantitative bearing fault diagnosis method based on the optimized Variational Modal Decomposition Lempel–Ziv complexity (VMD-LZC) was developed to fully extract the vibration characteristics and to quantitatively characterize the bearing faults under variable operating conditions. First, to compensate for the deficiency that the main parameters of the variational modal decomposition (VMD) have to be selected by human experience, a genetic algorithm (GA) is used to optimize the parameters of the VMD and adaptively determine the optimal parameters [*k*, *α*] of the bearing fault signal. Furthermore, the IMF components that contain the maximum fault information are selected for signal reconstruction based on the Kurtosis theory. The Lempel–Ziv index of the reconstructed signal is calculated and then weighted and summed to obtain the Lempel–Ziv composite index. The experimental results show that the proposed method is of high application value for the quantitative assessment and classification of bearing faults in turbine rolling bearings under various operating conditions such as mild and severe crack faults and variable loads.

## 1. Introduction

Micro-turbine rolling bearings, which are found in turbine coolers for aircraft environmental control systems, are often subjected to high temperatures, high pressures and frequent take-offs and stops [1,2]. Once a bearing is damaged, it will make the micro-turbine exhibit abnormal vibration and emit noise, make the aircraft ring control system unable to operate, or even, in serious cases, cause the destruction of the plane [3]. According to statistics, bearing failures account for about 30% of the total failures in aircraft environmental control systems. Therefore, accurate identification of bearing failures is essential for safe and continuous operation and predictive renewal of rolling bearings, as well as for condition monitoring and reliable operation of aircraft environmental control systems.

Numerous scholars have proposed Empirical Modal Decomposition (EMD) [4], Stochastic Resonance (SR) [5,6], Wavelet Packet Decomposition (WPD) [7,8], and other techniques for diagnosing the characteristics of bearing fault signals and achieved positive results. However, the above-mentioned techniques are based on the assumption of a linear system, and the analysis process is not adaptive, so they cannot accurately describe the inherent characteristics of the signal. Local Mean Decomposition (LMD) and Ensemble Empirical Modal Decomposition (EEMD) have been successfully used as self-adaptive time-frequency analysis methods to extract fault characteristics in nonlinear systems [9,10]. However, they all suffer from serious problems such as endpoint effects and modal confounding phenomena [11,12].

Variational Modal Decomposing (VMD) [13] is a novel time–frequency decomposition method that enables adequate characterization of nonlinear non-stationary signals generated by nonlinear systems [14]. It is a non-recursive conceptual framework with excellent convergence, strict implementation, constraint capability, and robustness under strong noise [15]. The VMD method has proven to be successful in fault diagnosis of bearing gears and rotating machinery [16]. For example, Li et al. [17] proposed a method to select sensitive IMF(s) of VMDs containing rich fault information based on FBE. A fault-information-guided VMD method is introduced by Qing et al. [18] for extracting the weak bearing repetitive transient. Li et al. [19] suggested a shock fault detection method for gearboxes based on combining VMD and coupled underdamped stochastic resonance to extract periodic pulse features. A novel VMD method is proposed by Jiang et al. [20] to extract accurately the weak damage features of rotating machines. Although this method has some advantages over other methods, the method has a serious shortcoming in that the number of modal decompositions *k* and the penalty factor α must be set in a predetermined, empirical manner. The choice of these two parameters determines whether the fault information can be extracted accurately. To reduce the possibility of undesirable consequences caused by artificially set parameters, the optimization of *k* and α in the VMD algorithm is required. Wang et al. [21] have presented an adaptive VMD method utilizing an Archimedean optimization algorithm. Miao et al. [22] proposed an improved parameter-adaptive VMD to extract all potential fault information, but the *k* values obtained with the optimized iterative algorithm are so unstable that they seriously affect the diagnostic performance.

In recent years, most industrial process monitoring and fault diagnosis has been concentrated on the detection and identification of faults. However, theoretical studies on the quantitative assessment of bearing failures under variable operating conditions have rarely been considered. Complexity is a powerful tool for characterizing the complexity of time series and is capable of quantitatively describing the state variation of a system [23,24]. It has been widely used in recent years to quantify the assessment of the severity of rotating equipment failures [25,26]. For example, Cui et al. [27] proposed a bearing FDI method based on a sparse diagram and Lempel–Ziv complexity (LZC) indicator, which has a better diagnostic effect than the traditional LZC method. Yin et al. [28] proposed an improved Lempel–Ziv method for bearing FDI based on symbol set approximation. Zhou et al. [29] proposed an improved LZC based on variable-step multi-scale analysis and an equivalent spatial partitioning strategy to fully exploit the vibration characteristics of rotating machinery. Dang et al. [30,31] proposed a novel fault severity evaluation method combining optimized multi-dictionary matching pursuit with LZC. Although certain progress has been made in extracting fault features of nonlinear systems, as mentioned above, the extracted features are prone to lose the intrinsic dynamic information of the system and cannot adequately characterize the nonlinear non-stationary signals.

On the one hand, it is important to determine how to obtain data containing both system state information and system dynamic information from the response signals of a turbine rolling-bearing system under variable speed and variable load conditions. It is also important to produce a comprehensive quantitative representation of the characteristics of these non-smooth, nonlinear, non-Gaussian stochastic signals. On the other hand, once the bearing is damaged in practical engineering, it will cause abnormal vibration and release noise in the micro-turbine. Powerful background noise, disturbances from abnormal pulses and vibrations from other internal components, and the effects of nonlinear factors such as manufacturing errors, assembly reasons, stiffness variations, friction, and damping make it difficult to quantitatively identify latent, inchoate signals characteristic of nonlinear systems at an early stage.

In response to the above challenges, the optimized VMD-LZC quantitative bearing fault diagnosis method is proposed to fully exploit the nonlinear characteristics and to achieve quantitative characterization of bearing faults under variable operating conditions. The process and its innovations are as follows:Introducing the Kurtosis theory in the optimized VMD, the IMF components that contain the most fault information are selected to reconstruct the signal. The method is independent of predetermined parameters and is capable of adaptive decomposition, which further improves the performance of the analysis of nonlinear and non-stationary signals generated by turbine systems and provides a basis for fully exploiting the characteristic expressions of the response signals of turbine systems.The Lempel–Ziv index of the reconstructed signal is calculated and then weighted and summed to obtain the Lempel–Ziv composite index. The Lempel–Ziv complexity index is used to evaluate the changes in the fault state of the bearing dynamics system. Thus, an innovative method for quantitative characterization of the dynamics of nonlinear turbine systems is proposed. This method can solve the problem of relying on the quantitative criteria of the current diagnostic technique to determine the fault with certain limitations and provide some reference for developing an automatic and scientific diagnostic technique in the future.Experimental turbine-bearing fault signals are processed by the proposed method. The results show that the proposed method can achieve quantitative evaluation and classification of faults for both mild and severe crack faults of turbine rolling bearings under various operating conditions such as variable load, and it has high application potential. Thus, it provides a new diagnostic philosophy for the quantitative diagnosis of turbine rolling-bearing faults.

## 2. Propaedeutics

### 2.1. Variational Modal Decomposition (VMD)

The VMD iteratively searches for the optimal solution of the variational model by the alternate direction method of multipliers, which can achieve the effective decomposition of the signal adaptively. The specific construction steps are as follows.

First, its one-sided spectrum is acquired by Hilbert transform, and its spectral expression is as follows:(1)$$\left(\delta \left(t\right)+\frac{j}{\pi t}\right)\times u_{k}\left(t\right)$$

Second, by mixing the individual mode-resolved signals, the pre-estimated center frequency $e^{-j\omega kt}$ is derived:(2)$$\left[\left(\delta \left(t\right)+\frac{j}{\pi t}\right)\times u_{k}\left(t\right)\right]e^{-j\omega k^{t}}$$

Finally, the bandwidth of the modal component is estimated by calculating the squared value of the time-gradient parameter $L_{2}$ of the demodulated signal.
(3)$$\|\partial_{t}{\left[\left(\delta \left(t\right)+\frac{j}{\pi t}\right)\times u_{k}\left(t\right)e^{-j\omega k^{t}}\right]\|}_{2}^{2}$$

Thus, the constructive equation of the variational model subject to constraints is as follows:(4)$$\underset{\left\{\mu_{k}\right\},\left\{\omega_{k}\right\}}{\min}\left\{\overset{k}{\underset{k=1}{\sum }}\|\partial_{t}{\left[\left(\delta \left(t\right)+\frac{j}{\pi t}\right)\times u_{k}\left(t\right)e^{-j\omega k^{t}}\right]\|}_{2}^{2}\right\}\;\text{s.t.}\;\;\sum_{k=1}^{k}u_{k}=f$$

In the above equation, $\delta \left(t\right)$ denotes the Dirichlet function, * denotes the convolution operation, $\left\{u_{k}\right\}$ = $\left\{u_{1},\cdot \cdot \cdot ,u_{k}\right\}$ denotes the set of *K* BLIMFs after VMD, $\left\{\omega_{k}\right\}$ = $\left\{\omega_{1},\cdot \cdot \cdot ,\omega_{k}\right\}$ denotes the union of the center frequencies of *K* modal components, and f denotes the input signal.

A quadratic penalty factor α and Lagrange multiplier operator $\lambda \left(t\right)$ are implemented to resolve the problems that may occur when Gaussian noise is present. The augmented Lagrange multiplier ζ is as follows.
(5)$$\zeta \left(\left\{\mu_{k}\right\},\left\{\omega_{k}\right\},\lambda \right)=\alpha \sum_{k=1}^{k}\|\partial_{t}{\left[\left(\delta \left(t\right)+\frac{j}{\pi t}\right)\times u_{k}\left(t\right)e^{-j\omega k^{t}}\right]\|}_{2}^{2}\;+\|f\left(t\right)-\sum_{k=1}^{k}u_{k}\left(t\right){\|}_{2}^{2}+<\lambda \left(t\right),f\left(t\right)-\sum_{k=1}^{k}\mu_{k}\left(t\right)>\;$$

In the above equation, α denotes the equilibrium coefficient.

The ADMM algorithm is used to iteratively update $u_{k}^{n+1}$, $\omega_{k}^{n+1}$, and $\lambda^{n+1}$. The saddle point in the extended Lagrangian representation is found. The update of modal $u_{k}^{n+1}$ can be equated to the solution to the minimization problem.
(6)$$u_{k}^{n+1}=\arg\underset{u_{k}\epsilon X}{\min}\left\{\alpha \sum_{k=1}^{k}\|\partial_{t}\left[\left(\delta \left(t\right)+\frac{j}{\pi t}\right)\times u_{k}\left(t\right)e^{-j\omega k^{t}}\right]{\|}_{2}^{2}+\|f\left(t\right)-\sum_{i}u_{i}\left(t\right)+{\frac{\lambda \left(t\right)}{2}\|}_{2}^{2}\right\}$$

Equation (6) is subjected to the Parseval/Plancherel Fourier isometric transform, which is transformed to the frequency domain, to obtain the updated expression for the *k*th modal.
(7)$${\hat{u}}_{k}^{n+1}\left(\omega \right)=\frac{\hat{f}\left(\omega \right)-\sum_{i<k}{\hat{u}}_{i}^{n+1}\left(\omega \right)-\sum_{i>k}{\hat{u}}_{i}^{n}\left(\omega \right)+\frac{\hat{\lambda }\left(\omega \right)}{2}}{1+2\tau {\left(\omega -\omega_{k}^{n}\right)}^{2}}$$

According to the same theory, the center frequency solution is transformed to the frequency to obtain the update to the center frequency as follows.
(8)$$\omega_{k}^{n+1}=\frac{\int_{0}^{\infty }\omega {\left|{\hat{u}}_{k}^{n+1}\left(\omega \right)\right|}^{2}d\omega }{\int_{0}^{\infty }{\left|{\hat{u}}_{k}^{n+1}\left(\omega \right)\right|}^{2}d\omega }$$

In Equation (8), the center frequency $\omega_{k}^{n}$ is the center of gravity of its corresponding modal function power spectrum ${\hat{u}}_{k}^{n+1}$(ω). The Fourier inversion is performed on the Werner-filtered signal of ${\hat{u}}_{k}\left(\omega \right)$ to obtain the modal $u_{k}$(*t*) in the time domain signal and to obtain the real part of it.

The optimal solution of the variational model is obtained while using ADMM for constraint, and finally, *k* modal components decomposed from the original signal are available.

### 2.2. Optimizing the Parameter Factors of VMD

The setting of parameters *k* and α is crucial for the VMD’s decomposition of the vibration signal to be effective. When a micro-turbine bearing is damaged, its vibration response is disturbed by strong background noise, abnormal pulses, and other internal component vibrations. Simultaneously, manufacturing errors, assembly reasons, stiffness variations, friction, damping, and many other nonlinear factors can also interfere. All of them cause the response signal in the micro-turbine bearing system to have the characteristics of a non-smooth, nonlinear, and non-Gaussian random signal. According to the principle of genetic algorithm (GA), GAs can effectively solve the global nonlinear optimization problem and find the optimal parameter combination required according to the maximum and minimum values of a fitness function as optimization criteria. Therefore, during parameter selection, to avoid the inaccuracy of human presetting, a GA is proposed to accurately obtain the optimal parameters of the VMD with the marginal spectral function as the criterion. The adaptive determination of the modal components’ *k* and penalty factor α of the vibration data of the rolling-bearing fault in the turbine is realized. This effectively overcomes the over- and under-decomposition of the signal due to the improper combination of [*k*, α] parameters and the weak fault characteristics being obscured by noise. The flow chart of the VMD based on GA optimization is shown in Figure 1.

### 2.3. Kurtosis

The Kurtosis value indicator is effective as a screening component for the periodic shock component of the vibration signal related to bearing fault, and it can be used for the screening of signal components to effectively reduce noise interference components. The Kurtosis value *k* is defined as follows:(9)$$k=\frac{E{\left(x-\mu \right)}^{4}}{\sigma^{4}}$$

In the above equation, *x* is the vibration signal to be analyzed, *μ* and *σ* characterize the mean and standard deviation of the vibration signal *x*(*t*), and *E* denotes the mathematical expectation. Since only a few of the IMFs among the original vibration fault signals of turbine rolling bearings obtained after optimized VMD processing are generally sensitive to fault information, this paper proposes to utilize the Kurtosis value to select the effective IMF components that exhibit rolling-bearing faults.

### 2.4. Lempel–Ziv Complexity (LZC)

Lempel–Ziv complexity has the merits of high computational efficiency and the absence of parameter selection [27]. Thus, LZC has been widely used to evaluate the complexity of vibration sequences for machinery fault diagnosis. The algorithm for the LZC is as follows in Figure 2.

Reconstruction sequence $\left\{x_{1},x_{2},\ldots ,x_{n}\right\}$
(10)$$s_{i}={\{}_{0,x_{i}<\bar{x}}^{1,x_{i}\geq \bar{x}}$$

In the equation, $\bar{x}=(x_{1},x_{2},\ldots ,x_{n})$, $S=(s_{1},s_{1},\ldots ,s_{r})$, r<n, *Q* = *S_r_*_+1_, forming the *SQ* string.

According to the algorithm described above, the complexity reflects the rate at which new patterns are generated as the length of the sequence grows, and the LZC metric quantifies the complexity of a finite time sequence. Therefore, the LZC can describe the changes occurring in the sequence and can be used as a characteristic parameter for the state of the system represented by the signal.

## 3. Diagnostic Model for Bearings in Micro-Turbines

The rolling bearings of micro-turbines are constantly in a complex working environment of high speed, high temperature, and high power. At the same time, their actual work is affected by the frequent starting and stopping of the turbine cooler of the aircraft ring control system, and the simple harmonic vibration and random vibration generated by the turbine are transmitted to the rolling bearing through the support structure. The vibration excitation long cycle effect will trigger friction wear accumulation and fatigue accumulation, prompting the bearing part’s surface contact deformation from elastic to plastic. In severe cases, such as pitting and spalling, bearing parts exhibit surface cracking faults. Additionally, the vibration signal fault mechanism of rolling-bearing faults is very intricate, and there is no clear quantitative characteristic index to describe the fault of the inner and outer race and cage of the rolling bearings in micro-turbines.

### 3.1. Diagnostic Process

The optimized VMD-LZC rolling-bearing-in-micro-turbine fault diagnosis method was designed to perform adaptive extraction of fault features as well as classification relying on complexity values. The specific flow chart of rolling-bearing-in-micro-turbine fault diagnosis is shown in Figure 3, where the main steps within the chart can be seen as follows.

(1)Signal acquisition: Simultaneous acquisition of vibration data using 3 acceleration sensors on the rolling-bearing-in-micro-turbine experimental bench. The measurement locations were bearing radial, bearing axial, and the experiment table housing, and a total of 30 types of sample data were collected.(2)Optimized parameters: An optimized VMD method was proposed to optimize the VMD by GA. The optimal parameters of VMD were obtained by GA based on marginal spectrum function, and the modal component number *k* value and penalty factor *α*, which is the value of vibration data of rolling-bearing-in-micro-turbine fault, were determined adaptively.(3)Signal processing: The optimized VMD is used to decompose the inner and outer race and cage fault vibration signals of rolling bearings. Based on the Kurtosis, the modal components of the bearing fault vibration signals that have been processed by the optimized VMD method are calculated, and the modal components containing more fault information are selected. This can effectively boost the performance of the model, thus further improving the segmentation accuracy of rolling-bearing faults in micro-turbines.(4)Complexity calculation: The modal components selected according to the Kurtosis are used for complexity calculation. The complexity can be used to extract the important nonlinear indicators in the rolling-bearing-in-micro-turbine fault signal and realize feature fusion. Thus, the rolling-bearing-in-micro-turbine fault data are comprehensively portrayed and the features of each dimension are finely processed, realizing the classification of rolling-bearing-in micro-turbine faults with optimized VMD-LZC model.(5)Quantitative diagnosis: Optimized VMD-LZC method and advanced Ensemble Empirical Modal Decomposition Lempel–Ziv complexity (EEMD-LZC) method, Empirical Modal Decomposition-Lempel–Ziv complexity (EMD-LZC) method, and conventional Lempel–Ziv complexity (LZC) method are the proposed 3 methods that were used to quantitatively diagnose multiple faults of rolling bearings in micro-turbines under severe fault, slight fault, and loading conditions.

### 3.2. Experimental Platform

The experimental platform of the rolling bearing of the micro-turbines is mainly composed of 2 parts: the rotor system and the data acquisition system [32,33,34]. The rolling bearing–rotor system experimental bench is shown in Figure 4. The rotor system is mainly composed of a drive system module, spindle system module, tooling and test piece module, and lubrication system module, which can simulate the mechanical characteristics of the actual environmental control system. The drive system module consists of an asynchronous motor with a rated speed of 2800 r/min and a rated power of 750 w. Three acceleration sensors are arranged in order from left to right on the housing of the rolling bearing in the turbine bearing–rotor system test bench. The sensors are used to test the complex vibration behavior generated by bearing failures in the turbine bearing–rotor system. The experiment was set up with a motor speed of 2400 r/min (operating frequency of 40 Hz) and a sampling frequency of 4000 Hz for the data acquisition system.

The main shaft system connects the motor and the test piece module. During the experiment, the rolling bearings under different operating conditions are placed in the test piece module, and the vibration signals are measured by an accelerometer with a sensitivity of 1.3 V/mm. Four common operating conditions of rolling-bearing health, inner race fault, outer race fault, and cage fault were selected for the experiment, in which both inner race fault and outer race fault were EDM cutting scratches.

An EDM cutting scratch of 0.5 mm is defined as a slight inner fault, and an EDM cutting scratch of 2 mm is defined as a severe inner fault. The physical diagram of bearing faults is shown in Figure 5.
(11)$$\;\;f_{c}=\frac{f_{r}}{2}\left(1-\frac{d}{D_{m}}\cos\alpha \right)=\frac{f_{bi}}{N_{b}}$$
(12)$$f_{be}=\frac{f_{r}}{2}N_{b}\left(1-\frac{d}{D_{m}}\cos\alpha \right)$$
(13)$$f_{bi}=\frac{f_{r}}{2}N_{b}\left(1+\frac{d}{D_{m}}\cos\alpha \right)$$
(14)$$f_{b}=\frac{f_{r}D_{m}}{2d}\left(1-{\left(\frac{d}{D_{m}}\cos\alpha \right)}^{2}\right)$$
where $D_{m}$ is the pitch diameter, d is the roller diameter, $N_{b}$ is the number of rollers, α is the ball contact angle, and $f_{c}$, $f_{bi}$, $f_{be}$, and $f_{b}$ are the cage, inner race, outer race, and roller fault frequencies, respectively.

### 3.3. Vibration Signals Analysis

The vibration signals of the rolling bearings in micro-turbines in the states of cage fault, inner race fault, and outer race fault are calculated by the method proposed in this paper. The vibration data of each type of fault are selected as 10 s and divided into 100 sample groups of 0.1 s each. Each sample group contains 400 vibration data points under the condition of 2400 r/min speed.

For each of the 4 different working conditions, 8 sets of data were collected, and each set of data was collected for 100 s. Taking the first set of data of each working condition as an example, the time–frequency domain signal processing method was used to process the signals of the four bearings, and the time domain waveform of the original vibration acceleration signal of the bearings is shown in Figure 6. The Fourier transform of the original vibration signal under each working condition is processed, and the spectrum is shown in Figure 7.

Figure 6 and Figure 7 demonstrate that the vibration signal collected under the healthy condition of rolling bearings in micro-turbines is relatively stable with small amplitude. When the inner race or outer race of the rolling bearing in a micro-turbine is faulty, the vibration amplitude of the fault signal increases significantly, and the impact amplitude shows a clear periodicity. In addition, when the rolling-bearing-in-micro-turbine cage fault occurs, the vibration amplitude is obviously reduced compared with the first two faults. However, compared with the healthy condition, it still increases, and the complexity of the signal increases.

## 4. Vibration Signal Decomposition

### 4.1. Ensemble Empirical Modal Decomposition (EEMD)

Ensemble empirical modal decomposition is an improved method based on Empirical modal decomposition (EMD). The EEMD steps are as follows:

White Gaussian noise *n_i_*(*t*) with a mean value of 0 and a constant amplitude standard deviation N is added to the original signal *x*(*t*) to obtain the signal *x_i_*(*t*) = (*I* = 1, 2, 3, …, *N*).
(15)$$x_{i}(t)=x(t)+n_{i}(t)$$

EMD is carried out on *x_i_*(*t*), and IMF components of *K* and a remainder *r_i_*(*t*) are obtained every time:(16)$$x_{i}(t)=\sum_{j=1}^{K}c_{ij}(t)+r_{i}(t)$$
where *c_ij_*(*t*) is the *j*th IMF obtained after adding white Gaussian noise for the *i* th time, *j* = 1, 2, 3, …, *K*.

Using the principle that the statistical series of uncorrelated random sequences has a mean value of 0, the IMF corresponding to the above steps is subjected to an overall averaging operation to eliminate the effect of multiple additions of Gaussian white noise on the true IMF and obtain the EEMD-decomposed IMF and the residual term *r*(*t*).
(17)$$c_{j}(t)=\frac{1}{N}\sum_{i=1}^{N}c_{ij}(t)$$
(18)$$r(t)=\frac{1}{N}\sum_{i=1}^{N}r_{i}(t)$$
where *c_j_*(*t*) is the *j*th IMF of the original signal after EEMD. IMF components of *K* and a residual term are obtained.
(19)$$x(t)=\sum_{j=1}^{K}c_{j}(t)+r(t)$$

The added white Gaussian noise cancels itself out with a sufficient number of repetitions, which can effectively eliminate the modal mixing during EMD.

With the EEMD method to decompose the rolling-bearing fault signal in micro-turbines, the EEMD’s effects on bearing health status, bearing cage fault, inner race fault, and outer race fault are shown in Figure 8.

As can be seen from Figure 8, the vibration signals of the rolling-bearing health state, bearing cage fault, and bearing outer race fault are decomposed into 10 modal components using the EEMD method, and the vibration signals of the bearing inner race are decomposed into 9 modal components. The modal components have high similarity to the original signal components in the time domain and can represent the overall characteristics of the original signal. Although they have the same modulation frequency as the original signal in the spectrogram, each modal component shows different degrees of modal overlapping and endpoint problems. Meanwhile, the EEMD method decomposes the components with the same frequency modulation, especially the modal overlap of IMF2–IMF4 components of each fault signal, which is relatively serious. Hence, the EEMD method has some drawbacks in processing the fault signals of turbine rolling bearings, as it cannot fully exploit the nonlinear characteristics of the fault signals of the bearing cage, inner race, and outer race. It is mainly attributed to the randomness, nonlinearity, and non-smoothness of the fault signal. Although the EEMD method adds an extra step of the denoising process compared to VMD in the decomposition, it adopts a non-recursive decomposition method and has the envelope estimation error, which will affect the decoupling accuracy to a certain extent.

### 4.2. Optimized Variational Modal Decomposition (VMD)

To address the problems of modal confounding and endpoint problems in the decomposition processing of rolling-bearing-in-micro-turbine fault signals by EEMD, this paper proposes to decouple and analyze the turbine rolling-bearing fault response signals by the optimized VMD. First, to overcome the drawback that the main parameters of VMD need to be selected based on human experience, a genetic algorithm is adopted to optimize the parameters of VMD and determine the optimal parameters [*k*, *α*] of the bearing fault signal adaptively. The optimization-seeking iteration curve of the optimized VMD method for rolling-bearing-in micro-turbine fault signal processing is shown in Figure 9, and the specific parameter values are shown in Table 1. Secondly, based on Section 4.1, the same rolling-bearing-in-micro-turbine fault signal data is processed. Finally, the adaptive decomposition processing of the rolling-bearing-in-micro-turbine fault signal is realized.

According to Figure 9 and Table 1, when the optimized VMD method is used to decompose the turbine rolling-bearing vibration signal, in order to make the values of the number of modal decomposition *k* and the penalty factor *α* more accurate, 50 iterations of finding the optimum and calculating the average value are required to obtain the best parameter [*k*, *α*] values. The most suitable parameter values for the vibration signal of the rolling-bearing health state of the turbine are [4, 1986], and the optimal parameter values for the three faults of the bearing cage, inner race, and outer race are [4, 1981], [4, 1892], and [4, 1984], respectively. The effect of optimized VMD on the 3 faults, namely rolling-bearing health, bearing cage, and inner race and outer race of the micro-turbine, is shown in Figure 10, respectively.

It is indicated in Figure 10 that the rolling-bearing-in-micro-turbine vibration signal can be effectively decomposed by applying the optimized VMD method. The 4 modal components are smooth, hierarchical, and independent subseries, which means that the method can effectively avoid the modal mixing problem in the EEMD process and can obtain better decomposition results. Moreover, the improved SNR indicates that the signal loss problem is eliminated successfully, providing a significant advantage for nonlinear signal processing. The feasibility of the optimal selection of decomposition layers and penalty factors for VMD based on a genetic algorithm is also verified, which reduces the subjective error of human factors. This is mainly attributed to the non-recursive decomposition of the optimized VMD in obtaining the intrinsic mode function (IMF), which not only effectively avoids the modal mixing and noise interference but also maximizes the retention of the original information in the signal and restores the original complexity of the signal.

### 4.3. Kurtosis Value of the IMF Component

Based on the Kurtosis theory described in Section 4.3, the Kurtosis value can be used to represent the probability density spikiness of the signal. According to the actual engineering statistics, a Kurtosis value coefficient greater than 3 indicates a bearing fault. First, the IMF modal component of the vibration signal of the rolling bearing of the turbine is calculated with Equation (9), and the Kurtosis value of the IMF component of the bearing vibration signal is shown in Figure 11.

Figure 11 confirms that when the rolling bearing of a turbine is in a healthy condition, the overall vibration signal obeys a normal distribution, and the Kurtosis value is approximately less than 3, within which the Kurtosis value of the IMF1 component is the largest, at 1.94. When a bearing cage fault occurs, the Kurtosis values of the four IMF components increase, among which the Kurtosis value of IMF3 is the largest, at 5.82. When bearing inner race or outer race fault occurs, the Kurtosis values of the four IMF components are the largest, at 5.83 and 6.66, respectively. If the Kurtosis value of the IMF component is greater than 3, it implies that the component contains a large number of vibration shock components, which most likely contain effective fault components. Therefore, these IMF components can be extracted and considered as valid fault vibration signals. In summary, the optimal signal component for the rolling-bearing cage fault signal of the turbine is IMF3, and the optimal signal component for the inner race and outer race fault signal is IMF2.

## 5. Analysis of Complexity

Since the turbine rolling-bearing system is a very sophisticated nonlinear dynamical system, the Lempel–Ziv complexity index [27], a complexity portrayal method in nonlinear dynamical systems, is introduced to quantitatively identify the different fault states of rolling bearings in micro-turbines. According to the algorithm presented in Section 2.4, the rolling-bearing-in-micro-turbine vibration signal is averaged and binarized, whereby data in the signal greater than or equal to the average value are replaced by 1, otherwise by 0. Then, a binary sequence consisting of 0 and 1 is obtained. The Lempel–Ziv normalized values of the vibration signal and its envelope signal are determined by calculation, then multiplied with the weight coefficients and added together to obtain the final Lempel–Ziv composite index.

### 5.1. Comparative Analysis of Traditional Lempel–Ziv Indicators

The average of the above index curves obtained from the 100 sample arrays was calculated to obtain the box line diagrams of the 4 types of bearing data under each type of method. The box plots of the bearing data obtained by two methods, traditional LZC and EMD-LZC, are shown in Figure 12 and Figure 13, respectively.

In Figure 14, the indicator curves obtained by the traditional LZC method directly applied to the vibration data of rolling bearings in micro-turbines are all mixed together. The fluctuations of the four complexity index curves of bearing health state, bearing cage, inner race, and outer race fault are substantial, and it is very difficult to discriminate the states of turbine rolling bearings. Meanwhile, it can be seen from Figure 12 that the difference between the mean value of the vibration data of the 3 fault states such as the bearing cage, the inner race, and the outer race fault and the healthy state is 0.06, 0.09, and 0.12, respectively, taking the mean value of the vibration data of the healthy state of the turbine as the standard. Because of its reliance on binary conversion, LZC tends to lose some valid information in the time series, and its noise immunity is not guaranteed. Moreover, the LZC index can only extract fault information at a single scale, which makes it difficult to fully explore fault characteristics. Therefore, the traditional LZC method is not sufficient for the quantitative diagnosis of rolling bearings in micro-turbines.

According to Figure 15, the index curves calculated by the EMD-LZC method for the vibration data of rolling bearings in micro-turbines have a certain degree of distinction; for instance, the distinction between the bearing’s health state and the three types of complexity index curves such as bearing cage, the inner race, and the outer race is relatively obvious. However, the complexity index curves of the three kinds of bearing fault states are still mixed more seriously, and it is difficult to distinguish the three kinds of fault states. Furthermore, according to Figure 13, the mean values of the vibration data of the bearing cage, inner race, and outer race are 0.06, 0.09, and 0.12, respectively. The main reason is that the rolling bearings in micro-turbines are affected by multiple factors such as manufacturing errors, which leads to the nonlinear and non-smooth vibration signal characteristics. Moreover, EMD itself lacks a strict mathematical theory, which can generate under-envelope, over-envelope, end-point effects, and modal mixing problems. Therefore, it is difficult to quantitatively characterize the faults of turbine rolling bearings with the EMD-LZC method.

### 5.2. Comparative Analysis of Severe Faults

Based on the same turbine rolling-bearing vibration signal with severe fault, the optimized VMD-LZC and EEMD-LZC methods are calculated to obtain and compare the Lempel–Ziv index curves of the rolling-bearing-in micro-turbine health status with the cage, inner race, and outer race fault status. The optimized VMD-LZC index curves are shown in Figure 16, and the EEMD-LZC index curves are shown in Figure 17.

As can be seen from Figure 16, the complexity index curves of the optimized VMD-LZC bearing fault diagnosis method are relatively significant in differentiation. The complexity index increases sequentially from the inner race fault to the cage fault, which can clearly identify the rolling-bearing-in-micro-turbine health status with three heavy crack failures. The complexity index can also be seen with the increase in the number of sample sets, and the fluctuation of the complexity index curves of both the inner race and outer race of the bearing is relatively minor. Meanwhile, it can be seen from Figure 18 that when the average value of the vibration data from the health state is taken as the standard value, the differences between the average value and the standard value of the vibration data of the three fault states of the bearing cage, inner race, and outer race are 0.21, 0.38, and 0.47, respectively. Evidently, the optimized VMD demonstrates a strong advantage in the processing of rolling-bearing-in-micro-turbine vibration signals, which means that it is effective in removing the strong background interference and can generate better decomposition results. The optimized VMD can not only decompose the narrowband signal and retain its adaptive decomposition characteristics but also avoid the drawback of its spline fitting, thus further improving the decomposition accuracy of the fault signal. Therefore, the effective diagnosis of the fault state of rolling bearings in micro-turbines is realized. As can be seen from Figure 17, the complexity index curves of the EEMD-LZC rolling-bearing-in-micro-turbine fault diagnosis method are less distinguished. However, the fluctuation of the complexity index curves of both faults is relatively dramatic. Additionally, the cage fault complexity index curve of the rolling bearings in micro-turbines is located in the middle of the inner and outer race fault complexity index curve, and the discrimination degree between them is quite low, so it is easy to misjudge the fault identification structure. Meanwhile, it can be seen from Figure 19 that when the average value of the vibration data of the health state is taken as the standard value, the difference between the average value and the standard value of the vibration data of the three fault states of the bearing cage, inner race, and outer race are 0.09, 0.17 and 0.22.

### 5.3. Comparative Analysis of Slight Faults

The slight crack fault signals of the same bearing were calculated with the proposed optimized VMD-LZC method and the state-of-art EEMD-LZC method, respectively. The Lempel–Ziv Complexity index curves of bearing health status are then compared and analyzed with the bearing cage, inner race, and outer race fault status. The index curves of the optimized VMD-LZC method are shown in Figure 20, and the index curves of the EEMD-LZC method are shown in Figure 21.

Figure 20 demonstrates that the optimized VMD-LZC method designed in this paper can distinguish the turbine rolling-bearing health clearly from the bearing cage, inner race, or outer race slight cracked fault states. The complexity indexes increase sequentially from bearing cage faults to bearing outer race faults, and the differentiation of complexity indexes between bearing inner race and cage faults gradually increases as the number of sample arrays increases. In addition, the differentiation of complexity indexes of the inner race and outer race faults is relatively small, within which the 20th complexity index and the 65th complexity index are noticeably high. Meanwhile, it can be seen from Figure 22 that when the average value of the vibration data from the turbine’s healthy state is used as the standard, the differences between the average value and the standard value of the vibration data of the three fault states of the bearing cage, inner race, and outer race are 0.23, 0.35, and 0.45, respectively. This indicates that the complexity index proposed in this paper can extract the deep-level features embedded in the signal and can quantitatively characterize the three light crack faults, which can realize the diagnosis of bearing faults.

Figure 21 shows that the EEMD-LZC method can roughly distinguish the inner race and cage faults of rolling bearings in micro-turbines. However, the complexity index curves of these 4 bearing states fluctuate substantially, especially since the first 50 complexity indexes have a serious overlapping crossover phenomenon. The bearing outer race fault complexity index curve is located between the other two fault complexity index curves. Meanwhile, according to Figure 23, when taking the average value of the vibration data from the turbine’s healthy state as the standard, the differences in the average value of the vibration data between the three fault states of the bearing and the healthy state are 0.36, 0.37 and 0.32, respectively. Therefore, the EEMD-LZC method cannot effectively identify three kinds of light crack faults in the rolling bearings in micro-turbines. This indicates that in practical engineering, the weak periodic pulse features in the rolling-bearing-in-micro-turbine fault signal are so faint as to be difficult to extract due to the high background noise. After EEMD low-pass filtering, although the influence of some noise on the LZC index is eliminated, some effective information about bearing faults is also removed. Therefore, there are limitations on the quantitative diagnosis of inner race, outer race, and cage faults of turbine rolling bearings by the EEMD-LZC method.

### 5.4. Analysis of Bearing Loading Conditions

For the vibration signal of the same turbine rolling bearing under loading conditions, the optimized VMD-LZC method and the advanced EEMD-LZC method were utilized for calculation. The obtained bearing health status is compared and analyzed with the LZC index curves of three fault states, namely cage, inner race, and outer race fault. The optimized VMD-LZC index curve is shown in Figure 24, and the EEMD-LZC index curve is shown in Figure 25.

Figure 24 demonstrates that the optimized VMD-LZC method has the capability to identify three types of crack faults in the cage, inner race, and outer race of the rolling bearings in micro-turbines under loading conditions because of the superior differentiation of the complexity indicators. As the number of sample arrays increases, the complexity values of bearing outer race and cage faults differ more and are more distinguished, while the 55th and 56th of them have relatively large bounces in the complexity index. Meanwhile, it can be seen from Figure 26 that when the average value of the vibration data from the turbine’s healthy state is taken as the standard, the differences in the average value of the vibration data between the three fault states of the bearing and the healthy state are 0.22, 0.33 and 0.42, respectively. This indicates that the proposed method is able to extract the deep fault characteristics from the signals of micro-turbine rolling bearings in service under long-term variable load conditions. It also illustrates that the optimized VMD process not only eliminates the influence of some noise in the signal on the Lempel–Ziv Complexity index but also retains the maximum fault information of the bearing. Thus, the quantitative characterization of bearing faults under loading conditions can be achieved.

Figure 25 shows that the vibration response under bearing loading conditions processed by the EEMD-LZC method cannot clearly distinguish between the cage, the inner race, and the outer race faults. The complexity index curves of these three types of bearing fault data are all relatively fluctuating, and there is a serious crossover mixture problem. Furthermore, it can be seen from Figure 27 that when the average value of the vibration data from the turbine’s healthy state is taken as the standard, the differences in the average value of the vibration data between the fault states of the bearing cage, inner race, and outer race and the healthy state are 0.30, 0.31, and 0.34, respectively. Because, when the faults of rolling bearings in micro-turbines are in a loaded environment, the monitored faults often appear in the form of multiple-faults coupling and are accompanied by a larger noise component and a lower signal-to-noise ratio, the useful information of the signal is submerged in the noisy environment. This makes it difficult for the EEMD-LZC method to objectively quantify the complexity of bearing-fault time series. Therefore, the error rate of the EEMD-LZC bearing fault diagnosis method is large, and it cannot effectively identify the inner and outer ring faults and bearing cage faults of rolling bearings in micro-turbines.

To verify the advantages of optimized variational modal decomposition in modal confusion and noise interference, the optimized VMD-LZC method and the advanced EEMD-LZC method, EMD-LZC method, and LZC method are compared and analyzed for multiple types of faults of rolling bearings in micro-turbines under severe fault, slight fault, and loading conditions. The results show that optimized VMD can overcome the difficulty of nonlinearity and non-smoothness of rolling-bearing fault signals. The optimized VMD-LZC method is significantly better in differentiating the bearing fault signals under various variable operating conditions; in other words, the optimized VMD-LZC method can effectively and quantitatively assess the damage degree of rolling bearings quickly. The proposed improved method has a better recognition effect compared with the current more advanced feature-extraction methods. The main reason is that the optimized VMD-LZC indicator can decompose the bearing vibration signal with impulsive and strong background noise characteristics more effectively and improve the classification accuracy of the bearing fault signal. The results of processing the measured vibration signals also prove the effectiveness and feasibility of the optimized VMD-LZC method.

In summary, because the turbine rolling bearing suffers from various nonlinear and complex factors, the identification features of the current method are insufficient to obtain the ability to quantitatively identify the fault. The optimized VMD-LZC method shows more sensitivity in distinguishing the different fault severity degrees from the measured signal. The breakthrough attempts from quantitative to qualitative approaches regarding the pattern recognition of rolling-bearing fault diagnosis are realized.

## 6. Conclusions

To achieve a more visual and accurate quantitative diagnosis of rolling bearings and to meet the essential need for automated maintenance decisions, this study creatively conceived an optimized VMD-LZC method for fault diagnosis of rolling bearings in micro-turbines. The following conclusions can be made from the analysis of experimental results:

The proposed optimized VMD-LZC method can accurately and efficiently distinguish between the normal state of a rolling bearing in a micro-turbine and the three fault states of the bearing cage, inner race, and outer race. It also uses the marginal spectral function as a criterion to optimize the setting of VMD parameters to improve the accuracy of bearing fault diagnosis, thus providing a novel direction with enormous potential for the quantitative diagnosis theory of bearing faults. It also provides a new research idea for automatic diagnosis by conducting quantitative criteria for bearing fault diagnosis.

Since optimized VMD can almost eliminate the noise component of the signal while retaining all the fault information of the bearing, it can reflect the state of the nonlinear dynamical system more sensitively. Therefore, the Lempel–Ziv index can accurately reflect the nonlinear dynamics of rolling bearings. The proposed index can still accurately and comprehensively explore the fault characteristics despite the noisy environment, thus providing a new theoretical basis for rolling-bearing-in-micro-turbine fault diagnosis.

The proposed method is experimentally proven to be more accurate and detailed than the advanced EEMD-LZC method, EMD-LZC method, and traditional LZC method in showing the best quantitative identification of bearing faults. Thus, the problem that the traditional Lempel–Ziv struggles to obtain deep information due to manufacturing errors, assembly reasons, stiffness changes, friction, damping, and other coupling linkage factors with correlation is solved. It also solves the problem of inaccurate calculation results of the EEMD-LZC method due to modal confusion and other factors. Subsequently, it provides a simplified and well-defined evaluation index for identifying the faults of the inner race, outer race, and cages of rolling bearings in micro-turbines. This study has a broad potential and great application value and is of great significance to the fault monitoring and diagnosis of electrical equipment.

## Figures and Tables

**Figure 1 sensors-23-04044-f001:**
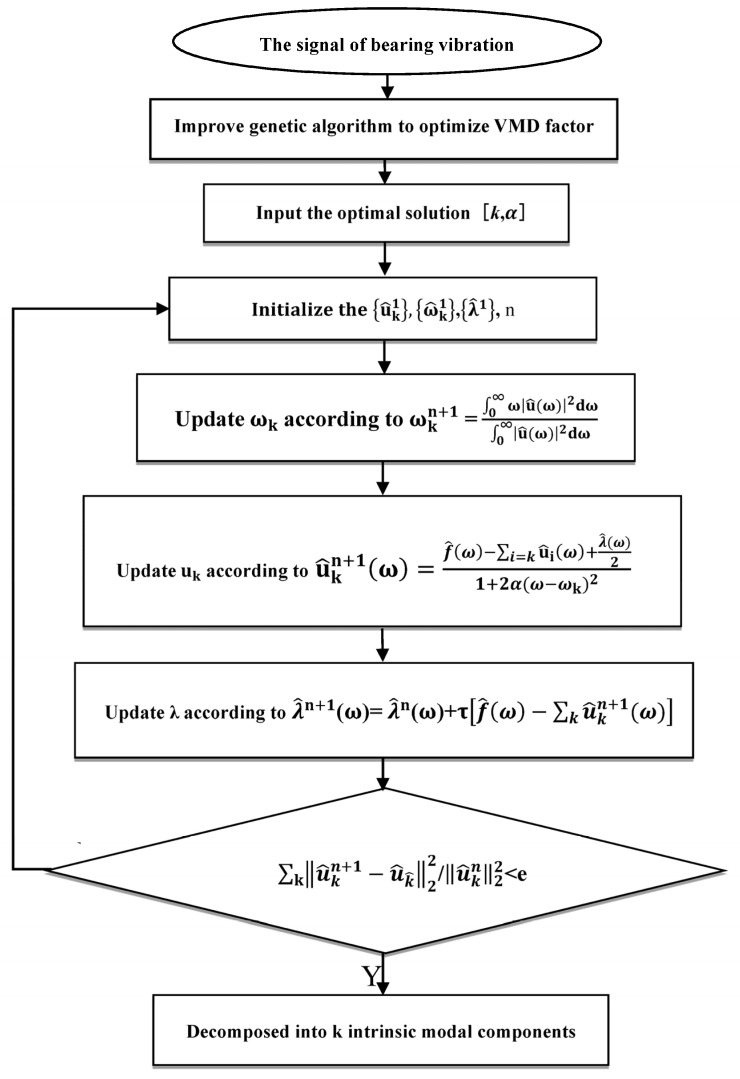
Parametric optimization flow chart of variational modal decomposition.

**Figure 2 sensors-23-04044-f002:**
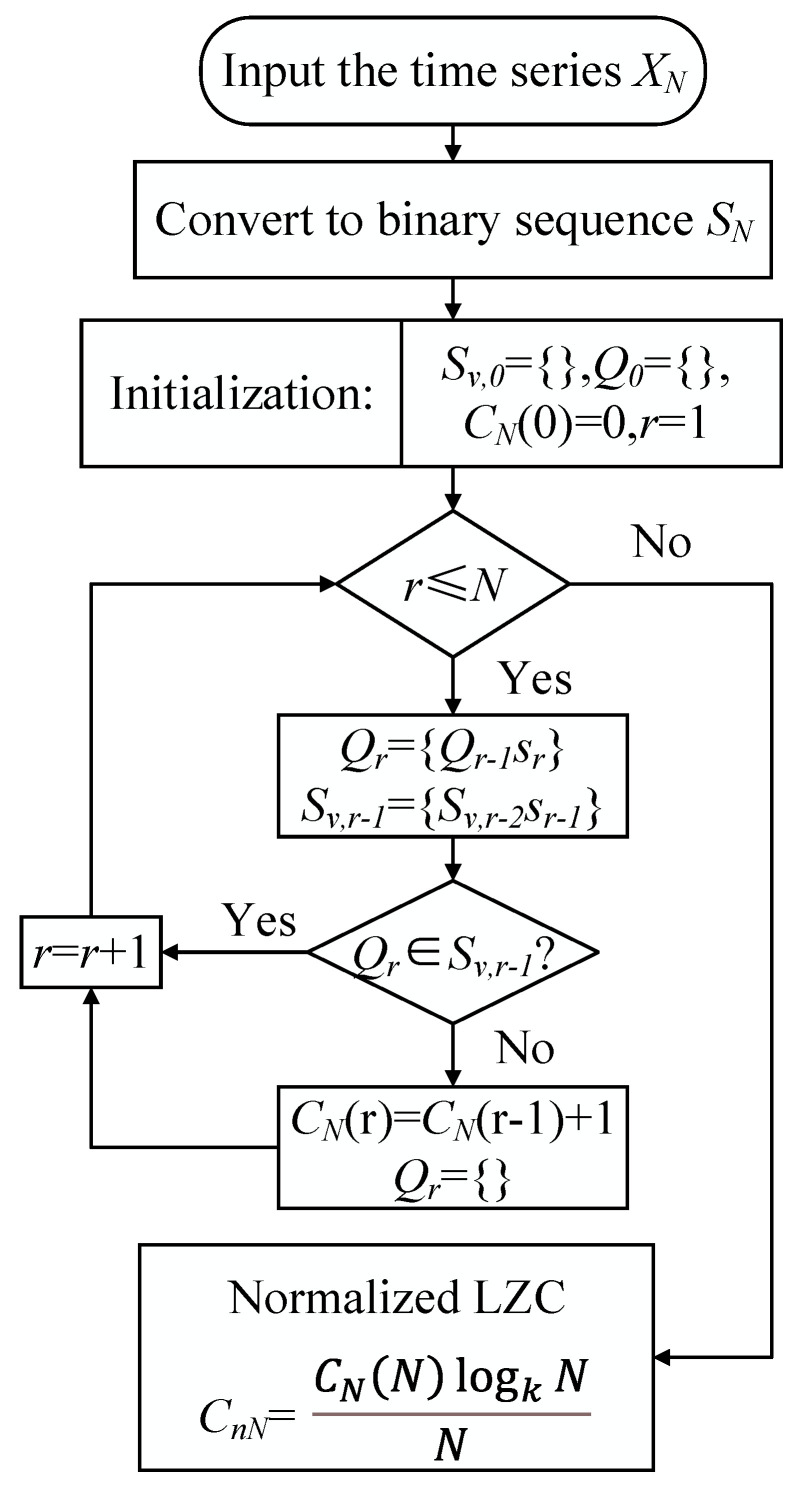
Lempel–Ziv complexity algorithm flow chart.

**Figure 3 sensors-23-04044-f003:**
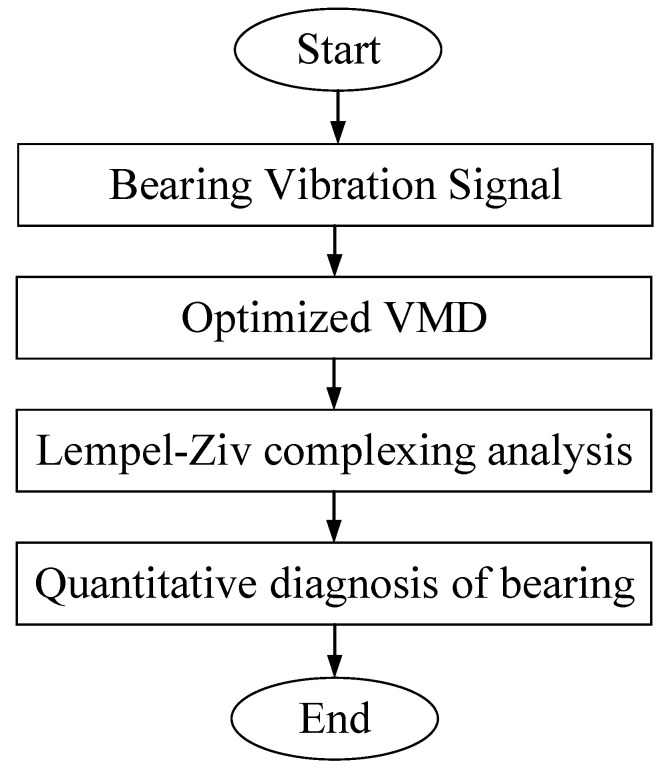
Shows an overall flow diagram of the proposed fault diagnosis method.

**Figure 4 sensors-23-04044-f004:**
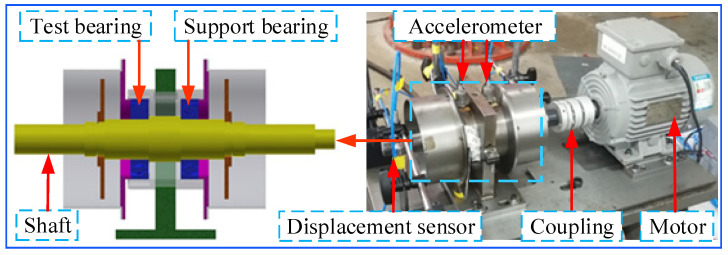
The turbine rolling bearing–rotor system experimental bench.

**Figure 5 sensors-23-04044-f005:**
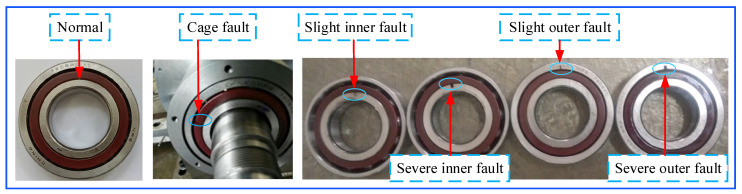
The rolling-bearing-in-micro-turbine fault forms.

**Figure 6 sensors-23-04044-f006:**
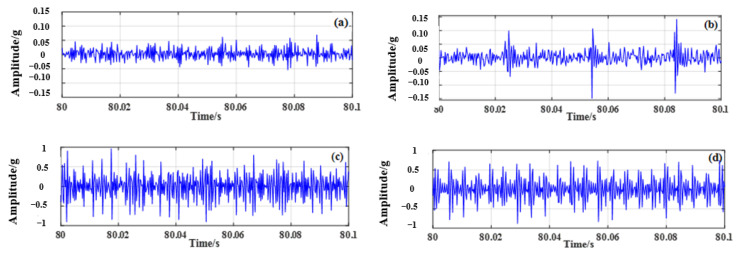
Time domain diagram of the vibration signal of the turbine’s rolling bearing under different operating conditions. (**a**) Health condition, (**b**) Cage failure, (**c**) Inner race failure, (**d**) Outer race failure.

**Figure 7 sensors-23-04044-f007:**
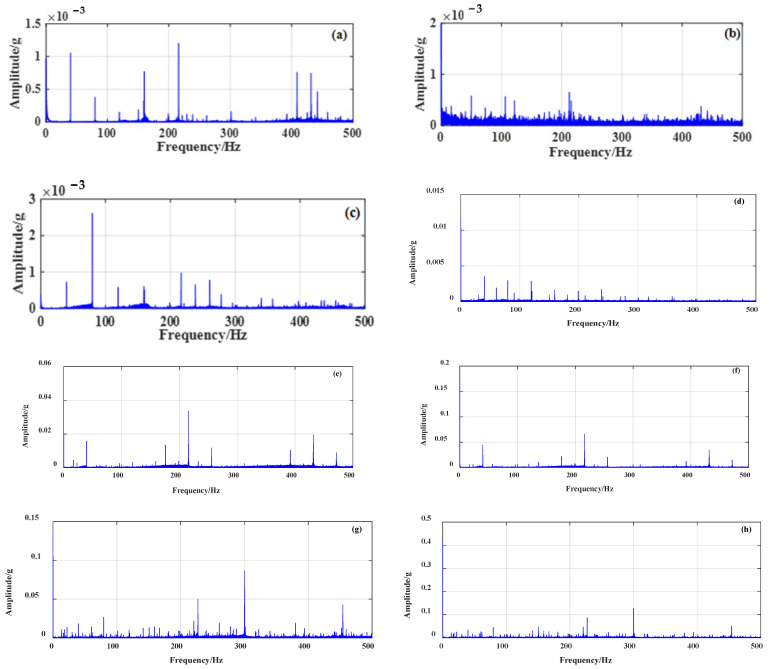
Spectrum and envelope spectrum of vibration signal under different working conditions. (**a**) Spectrogram of bearing health condition, (**b**) Envelope spectrum of bearing health condition, (**c**) Spectrogram of cage failure, (**d**) Envelope spectrum of cage failure, (**e**) Spectrogram of inner race failure, (**f**) Envelope spectrum of inner race failure, (**g**) Spectrogram of outer race failure, (**h**) Envelope spectrum of outer race failure.

**Figure 8 sensors-23-04044-f008:**
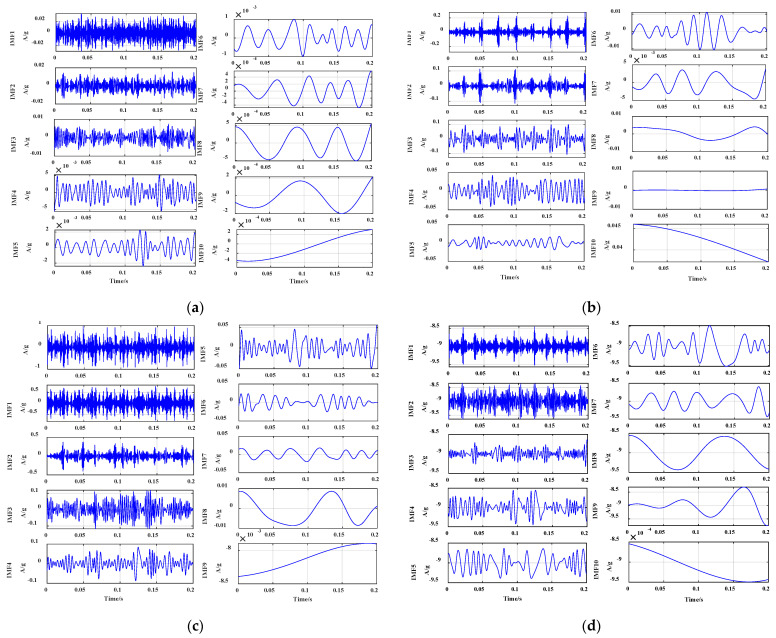
EEMD of the response signal of the rolling bearing of the micro-turbine. (**a**) IMF component of the vibration signal under healthy condition, (**b**) IMF component of the vibration signal under cage failure, (**c**) IMF component of the vibration signal under inner race failure, (**d**) IMF component of the vibration signal under outer race failure.

**Figure 9 sensors-23-04044-f009:**
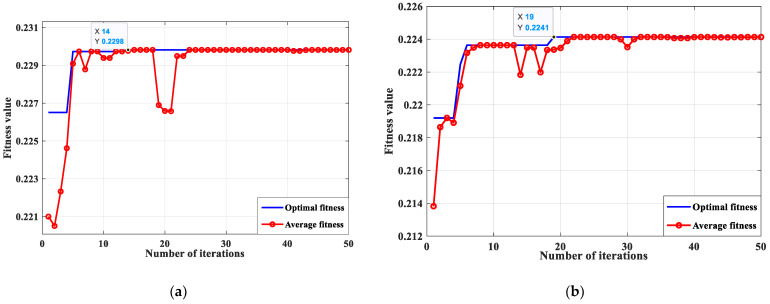
Optimization of VMD for turbine bearing vibration-signal-decomposition processing: optimization parameter curves. (**a**) Optimization curve for bearing healthy condition, (**b**) Optimization curve for cage failure, (**c**) Optimization curve for inner race failure, (**d**) Optimization curve for outer race failure.

**Figure 10 sensors-23-04044-f010:**
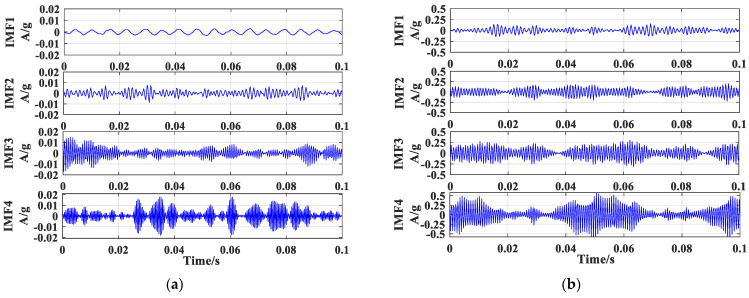
Decomposition of the response signal of the rolling bearing of the micro-turbine by the optimized VMD method. (**a**) IMF component of the vibration signal under healthy condition, (**b**) IMF component of the vibration signal under cage failure, (**c**) IMF component of the vibration signal under inner race fault, (**d**) IMF component of the vibration signal under outer race fault.

**Figure 11 sensors-23-04044-f011:**
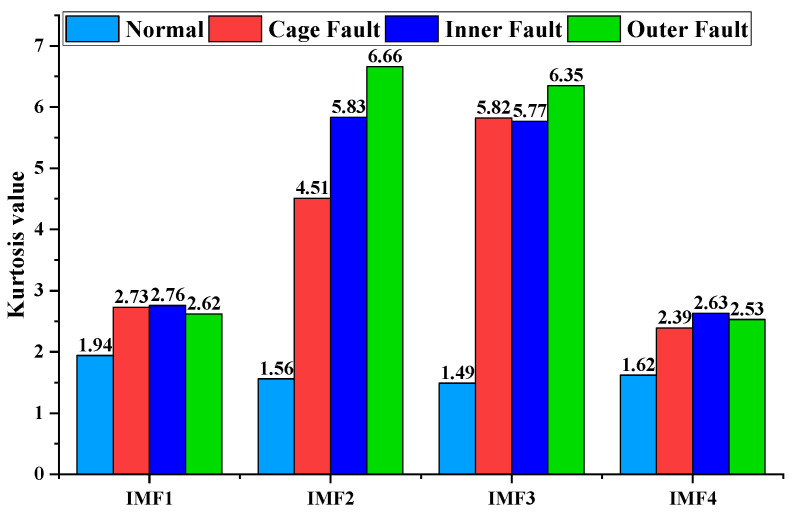
The bar chart of the Kurtosis value of the IMF component of the rolling-bearing vibration signal of the micro-turbine.

**Figure 12 sensors-23-04044-f012:**
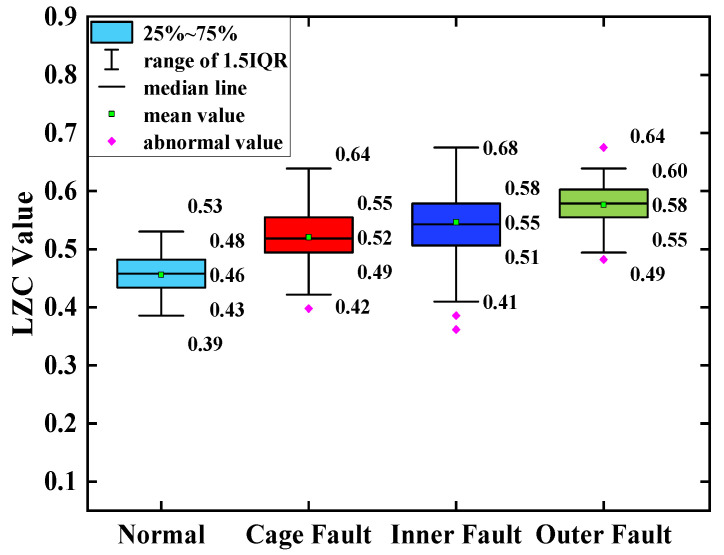
Box plot of traditional LZC indicator.

**Figure 13 sensors-23-04044-f013:**
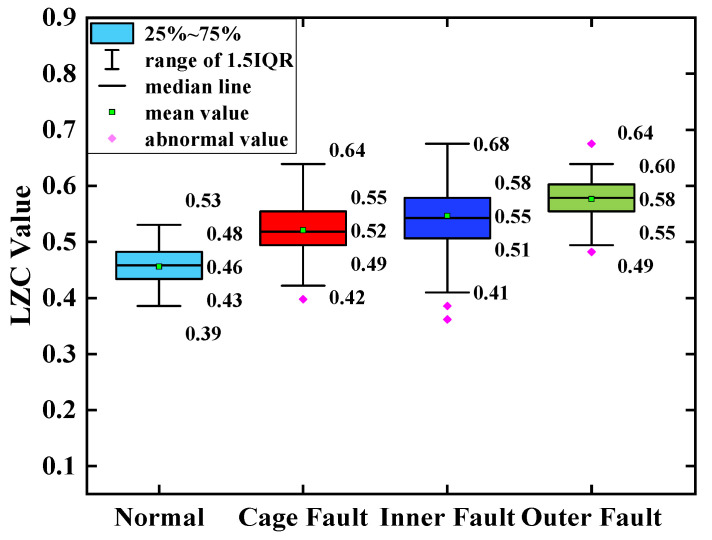
Box plot of EMD-LZC indicator.

**Figure 14 sensors-23-04044-f014:**
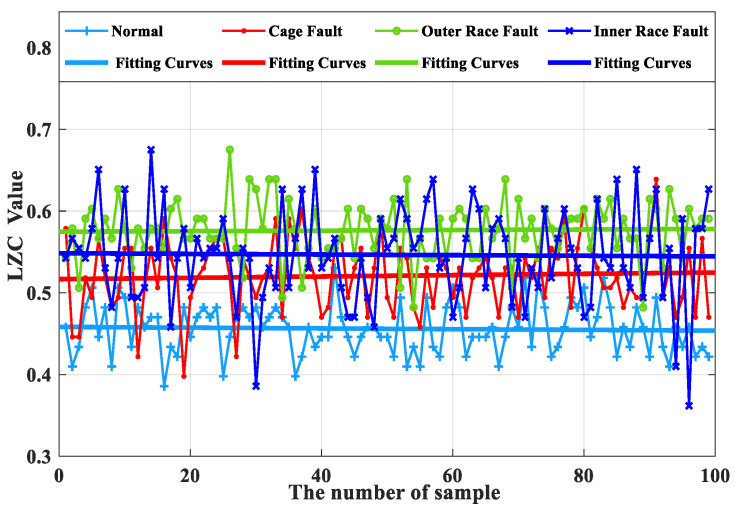
Traditional LZC index curve for 4 kinds of bearing fault data.

**Figure 15 sensors-23-04044-f015:**
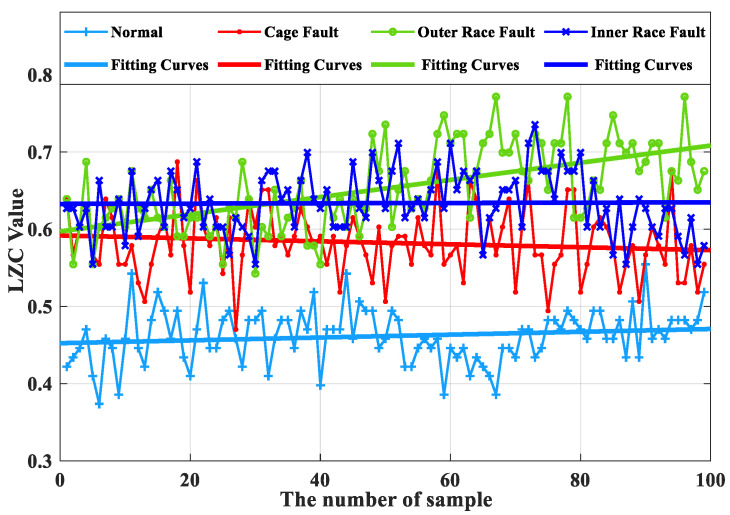
EMD-LZC index curve for 4 kinds of bearing fault data.

**Figure 16 sensors-23-04044-f016:**
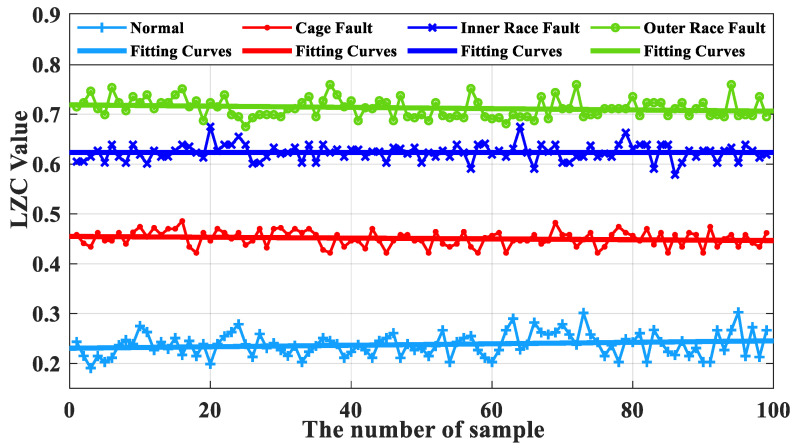
Comparison of optimized VMD-LZC index curves.

**Figure 17 sensors-23-04044-f017:**
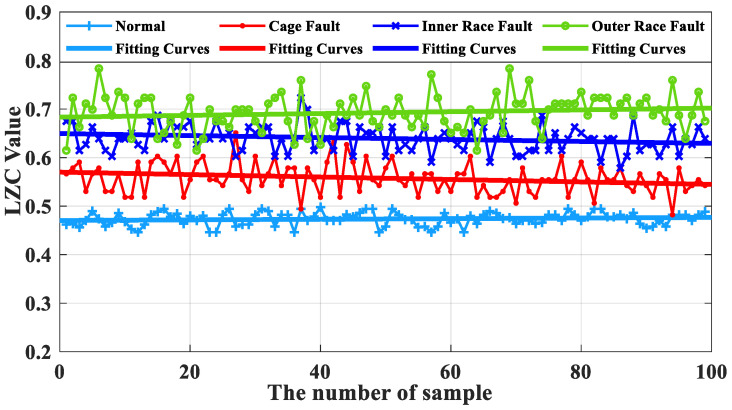
Comparison of EEMD-LZC index curves.

**Figure 18 sensors-23-04044-f018:**
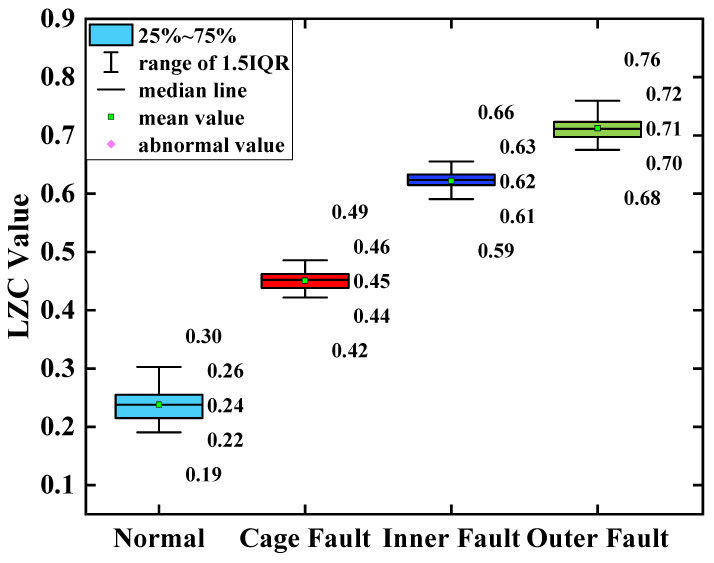
Box plot of optimized VMD-LZC index.

**Figure 19 sensors-23-04044-f019:**
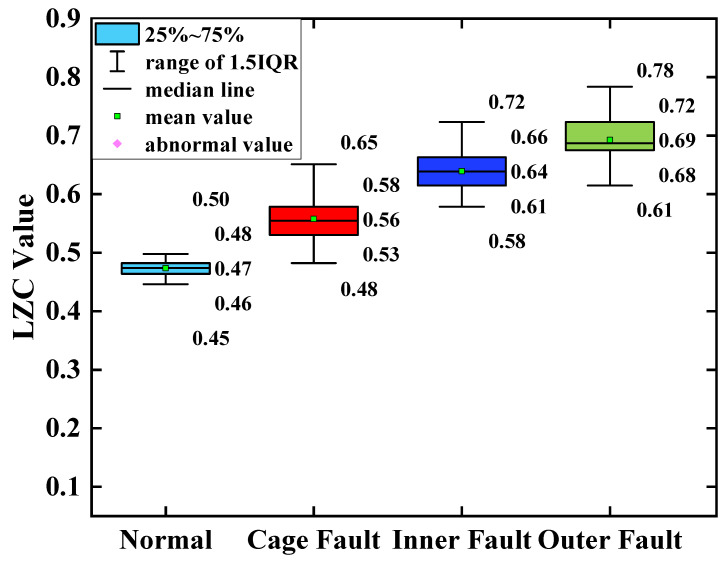
Box plot of EEMD-LZC index.

**Figure 20 sensors-23-04044-f020:**
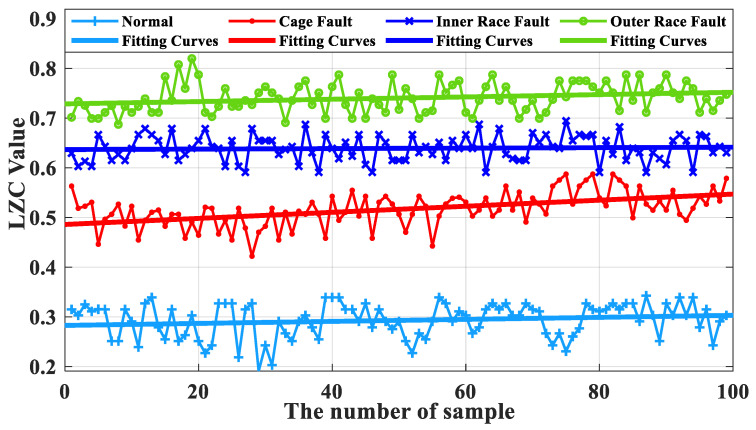
Comparison of optimized VMD-LZC index curves.

**Figure 21 sensors-23-04044-f021:**
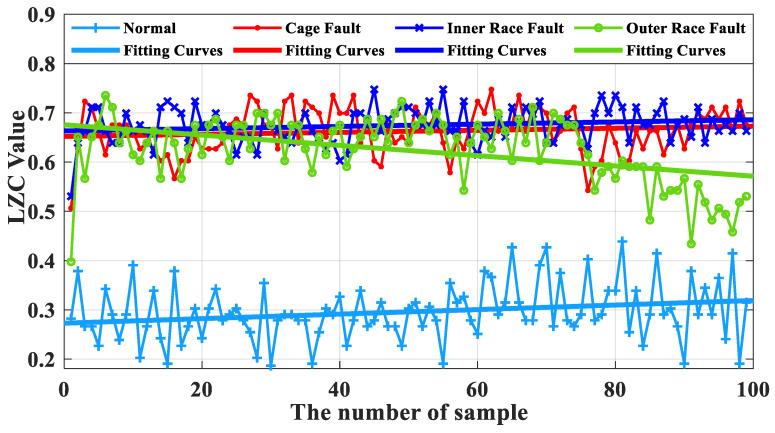
Comparison of EEMD-LZC index curves.

**Figure 22 sensors-23-04044-f022:**
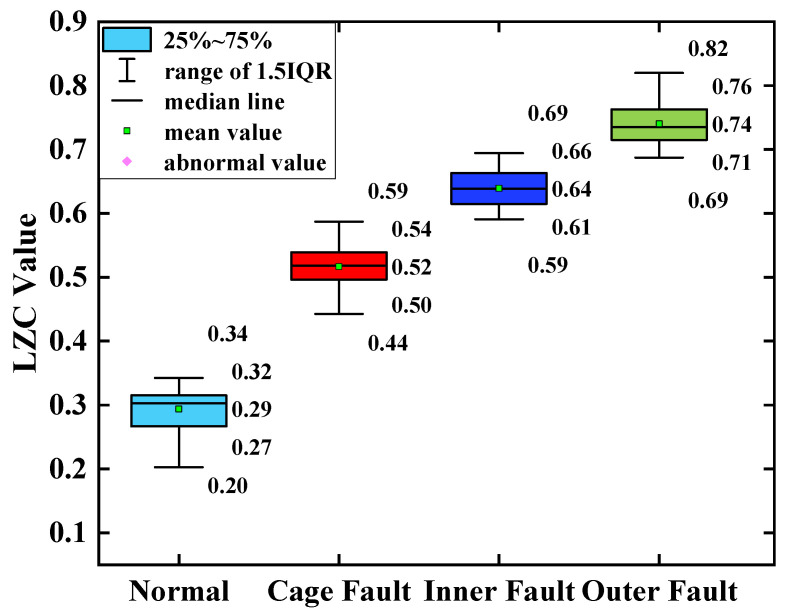
Box plot of optimized VMD-LZC index.

**Figure 23 sensors-23-04044-f023:**
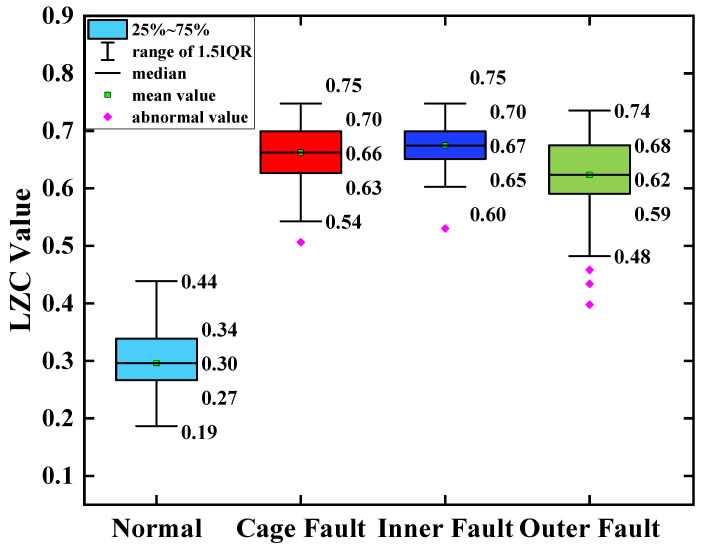
Box plot of EEMD-LZC index.

**Figure 24 sensors-23-04044-f024:**
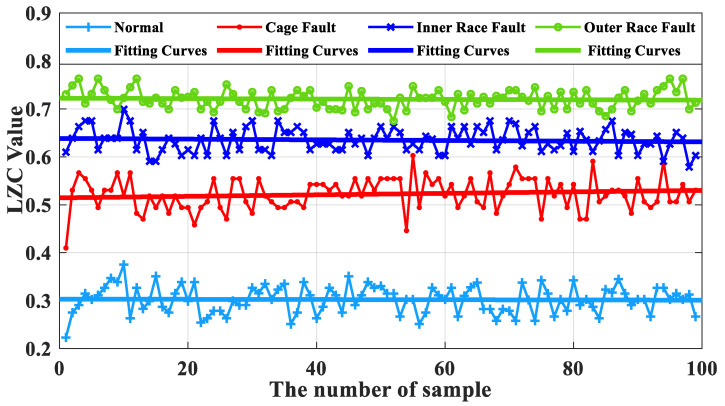
Comparison of optimized VMD-LZC index curves.

**Figure 25 sensors-23-04044-f025:**
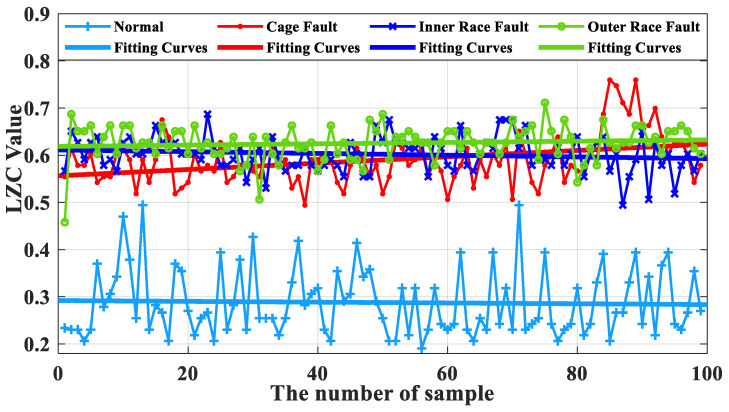
Comparison of EEMD-LZC index curves.

**Figure 26 sensors-23-04044-f026:**
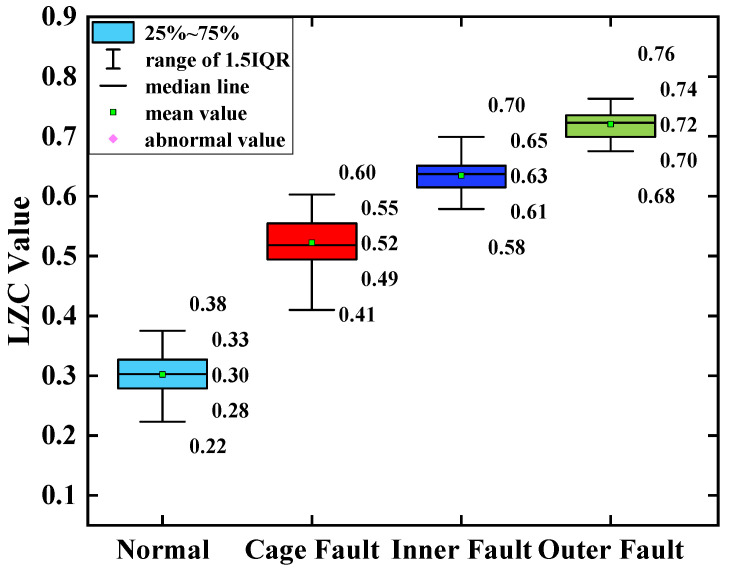
Box plot of optimized VMD-LZC index.

**Figure 27 sensors-23-04044-f027:**
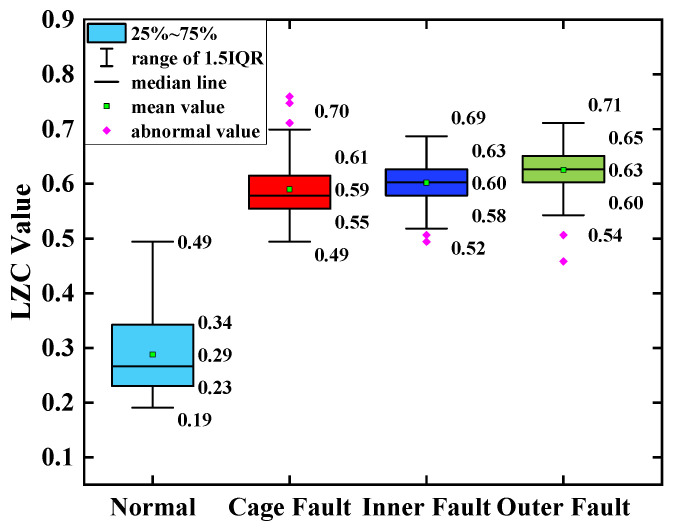
Box plot of EEMD-LZC index.

**Table 1 sensors-23-04044-t001:** VMD parameters of optimization.

Metric	Number of Iterations	Fitness Value	*K* Value	*α* Value
Normal	14	0.2298	4	1986
Cage Fault	19	0.2241	4	1981
Inner Fault	16	0.2138	4	1892
Outer Fault	27	0.2158	4	1984

## Data Availability

Not applicable.

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
