# Peer review of "Approach to the Quantitative Diagnosis of Rolling Bearings Based on Optimized VMD and Lempel–Ziv Complexity under Varying Conditions"

_sensors, 2023, doi:10.3390/s23084044_

Round 1

Reviewer 1 Report

In this paper, a diagnosis approach of rolling bearing is presented based on optimized VMD and Lempel–Ziv complexity. Generally speaking, much work has been done. However, some issues should be considered.

(1) The authors state that their diagnosis approach is quantitative. In my opinion, however, the paper lacks of the proof related to the quantitative. As we known, quantitative is of or relating to the describing or measuring of quantity. Thus, the features extracted with optimized VMD and Lempel–Ziv complexity for a fault should be stable in a certain value range or invariant to the changes of working condition. That does not mean that we just could use the extracted features or indicates to classify different faults under different working conditions, because many proposed features or indicates could do such thing very well. The quantitative feature means, in my opinion, this feature could measure the changes of the fault degrees (or there is a function/relationship between a quantitative feature and fault degrees) or the changes of the working conditions (or there is a function/relationship between a quantitative feature and various working speeds). Therefore, the features could classify the different faults under varying conditions are just qualitative instead of quantitative. I hope the authors could clarify this question clearly.

(2) Actually, VMD and Lempel–Ziv complexity has been used for fault diagnosis. Why the authors select the GA for optimization instead of other optimization methods?

(3) There are many features proposed to characterize the different faults under various working speeds. I think the authors could select some features for comparisons in the published related papers.

(4) There are some typos and language issues.

Author Response

Dear Reviewer

Thank you very much for your careful review of this manuscript, we have learned a lot from your valuable comments, we have carefully revised all according to your comments, please review again, if you have any other questions please contact us in time, thank you again.

Reviewer 2 Report

This paper creatively conceived an optimized VMD-Lempel-Ziv Complexity method for fault diagnosis of rolling bearings in turbines. The method is to provide a novel direction with enormous potential for the quantitative diagnosis theory of bearing faults. It also provides a new research idea of automatic diagnosis by conducting quantitative criteria for bearing fault diagnosis. There are still some writing errors in the article.

1、There are many grammatical errors in the icons in the papersuch as Fig.1,Fig.3 etc.

2、  The abbreviations of keywords used in the paper are not uniform and standardized. Such as Genetic Algorithm Variational Modal Decomposition LZC(GAVMD-LZC).

3、  The description of the experimental part of Section 3.2 in the paper is not standardized and clear. Such as the structure and working conditions in Fig.4 and Fig.5 are not fully expressed.

4、  What is the EEMD method ?

5、In Section 4.1 of the paper, how to get the signal decomposition needs to be supplemented by theoretical explanation.

6、  How to define the bearing mild crack fault and severe crack fault ?

7、The description in the paper is inner race failure, outer race failure, and cage failure, But in Fig. 5 the description is slight inner fault and Severe inner fault. Writing is not uniform, please modify.

Reviewer 3 Report

The paper describes a new quantitative bearing fault diagnosis method called the Genetic Algorithm Variational Modal Decomposition LZC (GAVMD-LZC) method. The authors claim that this method can fully extract the vibration characteristics of bearing faults under variable operating conditions and can quantitatively characterize bearing faults. The method is based on the Lempel-Ziv complexity (LZC) and uses a genetic algorithm to optimize the parameters of the variational modal decomposition (VMD) and adaptively determine the optimal parameters of the bearing fault signal. The authors also claim that the proposed method is effective for the quantitative assessment and classification of bearing faults in turbine rolling bearings under various operating conditions, such as mild and severe crack faults and variable loads. The paper presents a new method for bearing fault diagnosis that combines several techniques, including the Lempel-Ziv complexity, the variational modal decomposition, and a genetic algorithm. The experimental results provided in the paper show that the proposed method can effectively detect and classify bearing faults under different operating conditions. Therefore, the paper may be of interest to researchers and practitioners in the field of bearing fault diagnosis and may provide a new approach to improve the automation of maintenance decisions.

There are a few critical issues in the introduction:

1.The introduction could benefit from some restructuring to make it flow more logically and clearly. 

2.There are some grammatical errors and awkward phrasing that could be improved.

3. The introduction jumps between different techniques for analyzing bearing fault signals without giving a clear overview of each.

4.While the introduction touches on the importance of quantitative assessment of bearing failures, it doesn't fully explain why this is important or how it relates to the rest of the research.

5. There is a lack of context around the research problem, making it difficult to understand why this research is significant.

Proposed solution of the problem  aims to exploit the nonlinear characteristics of bearing faults and achieve quantitative characterization of these faults under variable operating conditions. The method introduces the Kurtosis theory in the optimized VMD to select IMF components that contain the most fault information for signal reconstruction. The Lempel-Ziv index is calculated to evaluate changes in the fault state of the bearing dynamics system and obtain a composite index that can be used for quantitative characterization of the dynamics of nonlinear turbine systems.

The proposed method is independent of predetermined parameters and is capable of adaptive decomposition, which improves the performance of the analysis of non-linear and non-stationary signals generated by turbine systems. The method can solve the problem of relying on the quantitative criteria in the current diagnostic technique to determine the fault with certain limitations and provide some reference for developing an automatic and scientific diagnostic technique in the future.

Experimental results using turbine-bearing fault signals show that the proposed method can achieve quantitative evaluation and classification of faults for both mild and severe crack faults of turbine rolling bearings under various operating conditions such as variable load, and has high application potential. Therefore, the proposed method provides a new diagnostic philosophy for the quantitative diagnosis of turbine rolling bearing faults.

Summarizing: The scientific results presented in this study demonstrate the effectiveness of an optimized VMD-Lempel-Ziv Complexity method for the fault diagnosis of turbine rolling bearings. The results show that the proposed method is superior to other advanced feature extraction methods in differentiating between bearing failure signals under various variable operating conditions. The GAVMD-Lempel-Ziv Complexity method can accurately and efficiently distinguish between the normal state of a rolling bearing and the three fault states of the bearing cage, inner race, and outer race.    The study concludes that the proposed method provides a novel direction with enormous potential for the quantitative diagnosis theory of bearing faults. The proposed method also provides a new research idea of automatic diagnosis by conducting quantitative criteria for bearing fault diagnosis. The GAVMD-Lempel-Ziv Complexity method can reflect the state of the nonlinear dynamical system more sensitively than other methods, and the Lempel-Ziv index can accurately reflect the nonlinear dynamics of rolling bearings.

Overall, the study is well-designed and the obtained results are clear and well-presented. The proposed method shows promising results and has the potential to provide a simplified and well-defined evaluation index for identifying the faults of the inner race, outer race, and cages of rolling bearings in turbines. However, it is important to note that the study was conducted on a specific type of machinery, and it may be necessary to further test the method on other types of machinery to determine its generalizability. Additionally, the limitations and assumptions of the proposed method should be carefully considered and addressed in future studies.

Final remark: Too much text is copied from [27]  without proper citation in all places.  (101 words / 

Lingli Cui, Beibei Li, Jianfeng Ma, Zhi Jin. "Quantitative trend fault diagnosis of a rolling bearing based on Sparsogram and Lempel-Ziv", Measurement, 2018  )

Author Response

Dear Reviewer,

Thank you very much for your careful review of this manuscript, we have learned a lot from your valuable comments, we have carefully revised all according to your comments, please review again, if you have any other questions please contact us in time, thank you again.
